# Green Synthesis and Antimicrobial Study on Functionalized Chestnut-Shell-Extract Ag Nanoparticles

**DOI:** 10.3390/antibiotics12020201

**Published:** 2023-01-18

**Authors:** Kai Shao, Jixiang Sun, Yamei Lin, Hongxin Zhi, Xitong Wang, Yujie Fu, Jiating Xu, Zhiguo Liu

**Affiliations:** 1Key Laboratory of Forest Plant Ecology, Ministry of Education, Northeast Forestry University, Harbin 150040, China; 2College of Chemistry, Chemical Engineering and Resource Utilization, Northeast Forestry University, Harbin 150040, China; 3Engineering Research Center of Forest Bio-Preparation, Ministry of Education, Northeast Forestry University, Harbin 150040, China; 4Heilongjiang Provincial Key Laboratory of Ecological Utilization of Forestry-Based Active Substances, Harbin 150040, China

**Keywords:** chestnut shell extract, hydrothermal synthesis, Ag nanoparticles, antimicrobial activity

## Abstract

The chestnut shell is usually discarded as agricultural waste and the random deposition of it can cause environmental problems. In this study, monodisperse crystalline Ag nanoparticles (AgNPs) were synthesized by a hydrothermal approach, in which the chestnut shell extract served as both reducing agent and stabilizer. The synthesized Ag nanoparticles were characterized by ultraviolet-visible (UV) spectrophotometry, transmission electron microscopy (TEM), Fourier transform infrared spectroscopy (FTIR), X-ray photoelectron spectroscopy (XPS) and X-ray diffraction (XRD) measurements. The TEM, XRD and XPS results revealed that the synthesized product was spherical Ag nanoparticles with a face-centered cubic crystal structure. The antimicrobial activity test indicated that the Ag nanoparticles modified by the chestnut shell extract had an obvious inhibitory effect on *Escherichia coli*, *Staphylococcus aureus* and *Candida albicans*. The measured MIC and MBC of functionalized chestnut-shell-extract AgNPs against *E. coli*, *S. aureus* and *C. albicans* is relatively low, which indicated that the present functionalized chestnut-shell-extract AgNPs are an efficient antimicrobial agent.

## 1. Introduction

Noble metal nanomaterials have attracted great attention in recent years and have been widely used in chemistry, energy, material science, food science, biomedicine and other fields [1,2]. Among the various noble metal nanomaterials, Ag nanomaterials have great potential to be developed as a safe and non-toxic inorganic antibacterial agent with some other superior properties [3]. It has been revealed that the Ag nanoparticles have a better antibacterial effect than that of traditional fungicides [4,5,6]. Therefore, the synthesis of Ag nanoparticles through different strategies has been explored in order to develop new kinds of antibacterial materials. The green synthesis of Ag nanomaterials with plant extracts is a promising approach to preparing eco-friendly and highly efficient antibacterial agents [1,3,7].

The green synthesis of AgNPs using plant extract as a reducing and stabilizing agent has proved to be promising strategy in the recent studies. The tuber extract of *Amorphophallus paeoniifolius* was utilized to green synthesize gold and silver nanoparticles (AuNPs and AgNPs) with antibacterial activity [8]. The callus extract of *Lithospermum erythrorhizon* can be served as a reducing and capping agent to prepare silver, gold, and bimetallic Ag/Au nanoparticles [9]. Green tea extract under visible light irradiation at room temperature was used to synthesize silver nanoparticles with photocatalytic degradation activity on methylene blue (MB) dye [10].

From the point view of resources, plants are widespread and renewable in nature, and contain many bioactive components, including polysaccharides, flavonoids and polyphenols [6]. Tens of thousands of tons of chestnuts are processed in the world and this activity produces large amounts of chestnut shells. The disposal of these chestnut shells as waste may cause some environmental problem. It has been revealed that chestnut shells contain many bioactive molecules including phenols, flavonoids, triterpenes, sugars, tannins and brown pigments with good water solubility [11,12]. Thus, chestnut shell is still a valuable source of bioactive molecules.

The brown pigment in chestnut shell especially has been demonstrated to have antioxidant, antibacterial and anti-inflammatory properties [13], which has attracted extensive attention in biomedical and industrial applications [6]. Therefore, the combination of chestnut brown pigment and Ag nanomaterials may have a synergistic effect for antibacterial activity. 

In this study, we attempt to use chestnut shell extract to synthesize Ag nanoparticles with superior antimicrobial activity. The chestnut shell extract was obtained by a microwave-assisted method and was then used for preparing the silver nanoparticles by hydrothermal synthesis [14]. The high pressure and temperature generated in the hydrothermal process can promote a redox reaction, which leads to the successful preparation of various novel nanomaterials [15,16,17,18]. The chestnut shell extract is believed to have a large quantity of flavonoids and polyphenols, which can be utilized as a source of reducing and stabilizing agents in the hydrothermal synthesis of AgNPs. One advantage of this study is that it utilizes waste plant material and avoids using harmful chemical agents in the green synthesis process. 

## 2. Materials and Methods

### 2.1. Materials

The chestnut materials in this study were collected in Qianxi, Hebei Province, China. The chemical reagents used included anhydrous ethanol (AR), AgNO_3_(GR), ammonia water (AR), hydrochloric acid (AR), nutritional agar (AR) and nutritious broth (AR). *Escherichia coli* (ATCC 8739), *Staphylococcus aureus* (ATCC 6538) and *Candida albicans* (ATCC 10231) were obtained from Heilongjiang Provincial Academy of Sciences Institute of Applied Microbiology (Harbin, China). Ultra-pure water (its resistivity was >18 MΩ cm) was used throughout this study.

### 2.2. Preparation of Chestnut Shell Extract

Chestnut shells were crushed with a pulverizer and the obtained powder was collected after passing through a 45 mesh sieve. A total of 5 g chestnut shell powder was placed into a round-bottom flask, and 100 mL 40% ethanol was added. The flask was then transferred into an ultrasonic microwave reactor. The extraction was maintained for 10 min under the condition of 360 W power, and then the extract was collected. The supernatant (~65 mL) of the extract was collected after centrifugation at 5000 r/min for 10 min. Following this, 20 mL of the supernatant was diluted to 40 mL with ultra-pure water and used for the following synthesis.

In a typical synthesis experiment, 40 mL of the diluted chestnut-shell-extract solution was mixed with 0.4 mL 5% AgNO_3_ solution. The solution’s pH was adjusted to 9.0 with the concentrated ammonia. After further stirring, 40 mL of solution was put into a Teflon-lined autoclave (50 mL), then the reactor placed into the oven and maintained at 140 °C for 4 h [14,17]. The produced solution was collected after the reaction for the following characterization. In this study, different synthesis conditions (pH ranging from 5.0 to 11.0 and ratio of the reactants ranging from 1:4 to 4:1 of the typical experimental conditions) have been investigated.

In order to compare with the results of functionalized chestnut-shell-extract Ag nanoparticles, Ag nanoparticles (AgNPs) were also synthesized by a sodium borohydride (NaBH_4_) reduction method [19], and can be regarded as nonfunctionalized AgNPs. The excess sodium borohydride can efficiently reduce silver nitrate to produce AgNPs. In a typical reaction, 10 mL of silver nitrate (1.0 mM) was gradually dropped into 30 mL of sodium borohydride solution (2.0 mM, chilled in an ice bath) over 3 min under rigorous stirring.

### 2.3. Characterization 

#### 2.3.1. Ultraviolet-Visible Absorption Spectra

The absorption spectra of the functionalized chestnut-shell-extract Ag colloid were determined by a UV2600 ultraviolet-visible spectrophotometer. The prepared functionalized chestnut-shell-extract Ag colloid was diluted 2 times, and the ultra-pure water was used as the blank control. At the same time, the ultraviolet spectrum of the chestnut shell pigment solution diluted 2 times was measured for comparison.

#### 2.3.2. Transmission Electron Microscope

A JEM-2100 transmission electron microscope at 200 kV was used for high resolution TEM observation. A total of 5 μL of functionalized chestnut-shell-extract Ag colloid was dropped on a carbon-coated copper mesh. The droplet was dried at room temperature prior to TEM imaging. 

#### 2.3.3. Dynamic Light Scattering (DLS) and Zeta Potential Analysis

Functionalized chestnut-shell-extract AgNPs were analyzed by particle size and zeta potential analyzer (MS2000/ZS90, Malvern, UK).

#### 2.3.4. Fourier Transform Infrared Spectroscopy

The infrared spectrum of the sample was measured by an IR Affinity-1 Fourier transform infrared spectrometer with a potassium bromide pressing method. The functionalized chestnut-shell-extract Ag nanoparticles were collected by centrifugation and dried at low temperature in the oven to obtain a solid powder. A total of 2 mg of functionalized chestnut-shell-extract Ag nanoparticles and 200 mg potassium bromide powder were fully ground and mixed to prepare the tablet for measurement. Chestnut shell extract in the absence of Ag material was also measured as a control.

#### 2.3.5. X-ray Diffractometer

The crystal structure of Ag nanoparticles was analyzed by an X-ray diffractometer (D/max-2200 V). It was equipped with CuK radiation (40 kV, 300 mA) of wavelength 0.154 nm operating at a scanning range 5–85°. The functionalized chestnut-shell-extract Ag nanoparticles for XRD measurement were collected by centrifugation.

#### 2.3.6. X-ray Photoelectron Spectroscopy

XPS data were collected by K-Alpha from Thermo scientific (East Grinstead, UK). A solid sample of Ag nanoparticles for XPS measurement were collected by centrifugation.

#### 2.3.7. LC-MS/MS Analysis

The chemical composition of the chestnut shell extract was analyzed by a Thermo Q Exactive™ focus (Thermo Scientific, Waltham, MA, USA) coupled with a Vanquish UHPLC system. The mass spectrometer was operated at a spray voltage of 3.0 kV in the positive-ion and negative-ion modes, respectively. The chestnut-shell-extract sample was detected using FullMS/dd-MS^2^ mode. The raw data were processed with Compound Discoverer 3.2 software (Thermo Scientific, Waltham, MA, USA).

#### 2.3.8. Antimicrobial Test

The Gram-negative bacteria *Escherichia coli*, Gram-positive bacteria *Staphylococcus aureus* and fungus *Candida albicans* were used for the antibacterial activity test using a standard disc-diffusion method [20]. *Escherichia coli* (*E. coli*), *Staphylococcus aureus* (*S. aureus*) and *Candida albicans* (*C. albicans*) were stored on nutrient medium at 4 °C. Strain activation was carried out before the experiment. Under aseptic conditions, 200 μL of three kinds of bacterial solutions (diluted to the logarithmic phase of 10^8^ CFU/mL) were smeared onto the agar medium plate. A total of 15 μL functionalized chestnut-shell-extract AgNPs (500 μg/mL) and control sample solution were added onto the filter paper with a diameter of 6 mm, and then the dried filter paper was placed onto the agar medium plate and incubated at 37 °C for 24 h. The growth-inhibition degree of the colonies was captured by a digital camera, and the size of the inhibition zone was measured. The experimental samples included functionalized chestnut-shell-extract Ag nanoparticles, chestnut-shell-extract solution, AgNO_3_ solution and (0.01%) penicillin solution as a control sample.

The minimum inhibitory concentration (MIC) and minimum bactericidal concentration (MBC) of functionalized chestnut-shell-extract AgNPs against *E. coli*, *S. aureus* and *C. albicans* were measured with the double dilution method [20]. The original silver colloid solution (5 μg/mL) was gradually diluted to 0.13 μg/mL by the nutrient solution. A total of 100 μL of dilute solution was transferred into the 96 well microtiter plate. A total of 100 μL of fresh bacteria suspension (1 × 10^6^ CFU/mL) was added into the wells. The prepared plate was cultured at 37 °C (*E. coli* and *S. aureus*) and 28 °C (*C. albicans*) for 24 h. After incubation, 10 μL of the incubated solution was re-inoculated on the agar plate for 24 h to check the bacterial growth.

## 3. Results and Discussion

### 3.1. Characterization of Functionalized Chestnut-Shell-Extract Ag Nanomaterials

Figure 1a shows the UV-Vis absorption spectrum of the functionalized chestnut-shell-extract Ag colloid and the chestnut shell extract in absence of any Ag component. An obvious absorption peak at 276 nm present for the chestnut-shell-extract solution (line a). It can be ascribed to the proanthocyanidins in the extract [21]. The broad absorption band from 300 to 800 nm was attributed to the ellagitannins in the extract. After the hydrothermal process, a sharp band peaking at 413 nm appeared (line b). It belongs to the surface plasmon resonance band (PRB) of the Ag nanoparticles, which has been verified by the recent studies [22,23,24]. The single absorption band in Figure 1a also shown that the synthesized products were mainly spherical nanoparticles and they were well separated according to the Mie’s theory [25].

The FTIR spectra of the functionalized chestnut-shell-extract Ag nanoparticles and chestnut shell extract were shown in Figure 1b. As for the FTIR spectrum of chestnut shell extract (line a), the broad absorption band peaked at 3265 cm^−1^ was ascribed to the stretching vibration of O-H (phenol) and the effects of the formation of a hydrogen bond. The absorption band peaked at 1608 cm^−1^ was attributed to the stretching of C=C group of the aromatic ring. The absorption band at 1446 cm^−1^ was due to the O-H in-plane deformation and the absorption band at 819 cm^−1^ resulted from the out-of-plane bending of the aromatic C-H bond [26]. In the FTIR spectrum of functionalized chestnut-shell-extract Ag nanoparticles (line b), all characteristic absorption bands are very similar to that of the chestnut shell extract. These results indicated that the chestnut shell extract was associated with the Ag nanoparticle. In comparison with FTIR spectrum of chestnut shell extract, the absorption band corresponding to the stretching vibration of O-H shift to 3167 cm^−1^ and the absorption band according to the O-H in-plane deformation shift to 1429 cm^−1^ as shown in Figure 1b (line b). The shift of O-H vibration both in stretching and deformation indicated that there are strong interactions between phenol components in the chestnut shell extract with the Ag atom on the nanoparticle surface.

Figure 2a shows the low-magnification TEM image of the as-synthesized functionalized chestnut-shell-extract Ag nanoparticles. It can be seen that the obtained products are small spherical nanoparticles. The enlarged TEM image shown in Figure 2b indicated the size of these Ag nanoparticles is relatively uniform. The size distribution of Ag nanoparticles is shown in the inset in Figure 2b. The average diameter of these Ag nanoparticles is 8.82 ± 1.1 nm (*n* = 100). Figure 2c shows the HR-TEM image of one typical spherical Ag nanoparticle. The lattice fringe spacing in Figure 2c is 0.238 nm, which can be attributed to the Ag (111) plane according to the recent literature [27]. This result indicated that the obtained nanoparticles had a clear crystalline Ag structure.

The size control of nanocrystals has attracted considerable attention due to its potential applications in nanotechnology [28]. In this study, different synthesis conditions (pH and ratio of the reactants) have been investigated. All primary TEM results indicated that the size of the synthesized Ag nanoparticles is approximately 10 nm at pHs from 7.0 to 11.0 as well as with different ratios of the reactants. This result is quite different to that of the classic citrate reduction method [28]. Nevertheless, the size of the functionalized chestnut-shell-extract Ag nanoparticles can be enlarged by the seed-growth method. A total of 20 mL of the as-synthesized Ag nanoparticles produced at 140 °C served as a seed for the synthetic solution and then maintained at 140 °C for 4 h.

Figure 2d shows the representative TEM image of the Ag nanoparticles by the seed-growth method. The observed Ag nanoparticles are still uniform. The average size is 14.4 ± 1.4 nm (*n* = 100). Figure 2e shows the representative TEM image of the functionalized chestnut-shell-extract Ag nanoparticles obtained at a pH of 7.0; the average diameter of these Ag nanoparticles is 11.2 ± 1.8 nm (*n* = 100). Figure 2f shows the typical TEM image of the functionalized chestnut-shell-extract Ag nanoparticles using twofold the amount of AgNO_3_ to that in the typical experiment; the average diameter of the Ag nanoparticles is 14.2 ± 2.7 nm (*n* = 100). 

The size and zeta potential of functionalized chestnut-shell-extract AgNPs were further analyzed by particle size and zeta potential analyzer. Figure 3 shows the size distribution of the functionalized chestnut-shell-extract AgNPs. The average diameter is 17.2 ± 5.0 nm, which is larger than TEM size. This can be ascribed to the DLS size measurement containing the hydrodynamic diameter of the chestnut-shell-extract layer attached on the surface of the Ag nanoparticle while TEM measurement cannot detect the chestnut-shell-extract layer. The average zeta potential of functionalized chestnut-shell-extract AgNPs is −22.0 ± 5.3 mV. The relatively high negative charge is advantageous for maintaining their stability.

The crystalline structures of the Ag nanoparticles were further confirmed by XRD measurement. Figure 4a shows the XRD of the as-synthesized functionalized chestnut-shell-extract Ag nanoparticles. The diffraction peaks at 38°, 45°, 65°, 78°and 82°correspond to (111), (200), (220), (311) and (222) of the face-centered cubic (fcc) Ag crystal [27,29]. The calculated crystal size is 7.80 nm according to the Scherrer equation based on the width of the (111) peak. This result is close to that of TEM measurement.

The element composition and surface chemical state of functionalized chestnut-shell-extract AgNPs were determined by XPS measurement as shown in Figure 4b. C, O and Ag element have been identified in Figure 4b. The C and O element can be attributed to the flavonoids and polyphenols in the chestnut shell extract. The two peaks at the binding energy of 367.9 eV and 373.9 eV in the XPS spectrum in Figure 4c are attributed to the Ag 3d_5/2_ and Ag 3d_3/2_ signals, respectively, and the potential difference is 6 eV, which is consistent with the standard position of zero valence Ag in the XPS spectrum [30]. This result is also in accordance with the XRD measurement, and confirmed that the surface of the nanoparticle is made up of metallic Ag at zero valent states.

### 3.2. Antibacterial Activity of Functionalized Chestnut-Shell-Extract Ag Nanoparticles

The antibacterial activity of the functionalized chestnut-shell-extract Ag nanoparticles was tested using a standard disc-diffusion method. The chestnut extract, AgNO_3_ solution and penicillin were also investigated for comparison. Figure 5 shows the photograph of the zone of inhibition of the functionalized chestnut-shell-extract Ag nanoparticles, the chestnut shell extract, AgNO_3_ solution and penicillin against (a) *Escherichia coli* (b) *Candida albicans* (c) *Staphylococcus aureus*. It can be seen that the functionalized chestnut-shell-extract Ag nanoparticles have bactericidal effects on all tested bacteria and fungi. Especially for *Candida albicans,* the synthesized Ag nanoparticles showed better antifungal activity than the control samples (the chestnut shell extract and AgNO_3_ solution). It should be noted that the chestnut shell extract and AgNO_3_ solution also have antibacterial activity. Their antibacterial activity differs among these tested bacteria and fungi. The antibacterial activity of the AgNO_3_ solution is derived from the Ag ions, which can disrupt the cell membranes, enzyme and nucleic acids of the bacterial. The antibacterial activity of the chestnut shell extract can be ascribed to the abundant flavonoid and polyphenol components, which can interrupt the membranes and proteins of bacteria by hydrogen bonding and non-covalent interaction. 

The measured MIC and MBC of functionalized chestnut-shell-extract AgNPs and control samples against *E. coli*, *S. aureus* and *C. albicans* are given in Table 1. The MIC and MBC of functionalized chestnut-shell-extract AgNPs are lower than those of AgNPs produced by the NaBH4 reduction method. It is indicated that the antimicrobial efficiency of the functionalized chestnut-shell-extract AgNPs is also lower than that of penicillin, according to the MIC and MBC values in Table 1. It has been noted that the MIC and MBC of functionalized chestnut-shell-extract AgNPs against *E. coli, S. aureus* and *C. albicans* are lower those of the biogenic AgNPs in a recent study [31,32].

Many factors, including size distribution, morphology, surface chemical composition, surface charge and capping agents, can significantly influence the antimicrobial activity of inorganic nanoparticles [33,34,35,36]. For instance, the comparison of plant-mediated fabricated AgNPs with commercial AgNPs indicated that there is significantly different antibacterial activity with minimum inhibitory concentration (MIC) of 4 and 8 µg/mL against *S. aureus*, respectively [33]. Therefore, the surface capping of chestnut shell extract on Ag nanoparticles may contribute to their antimicrobial activity. 

The broad-spectrum and relative strong bactericidal effect of the functionalized chestnut-shell-extract Ag nanoparticles can be explained as follows. It has been verified that the Ag nanoparticles gradually release Ag ions from the particle surface after oxidation by the dissolved O_2_ in the natural environment [37]. The produced Ag ions can denature the membrane proteins on the cell wall and enzymes in a cell, and disrupt the cell-wall structure of microbes, inducing the leakage of cytoplasm components. Another possible way for this to occur is that Ag atoms on the surface of the nanoparticle directly interact with the sulfur- and phosphorus-containing membrane protein and enzymes in the bacterial cell, inducing enzyme inactivation when the Ag nanoparticle attaches to the bacterial surface or endocytoses into the bacteria. The possible antimicrobial mechanism of the chestnut-shell-extract AgNPs is schematically illustrated in Figure 6.

### 3.3. Formation Mechanism of Ag Nanoparticles

The formation process of Ag nanoparticles in this study can be regarded as a typical redox reaction wherein the phenolic, phenolic acid molecules in the chestnut shell extract serve as the reducing agent and protect agent and Ag ions as the oxidant. The chemical composition of the chestnut shell extract has been investigated recently [38]. It is revealed that the chestnut shell extract contained the highest number of phenolic molecules and phenolic acids, such as gallic acid, catechin and epicatechin. Therefore, these phenolic molecules and phenolic acid play an important role in reducing Ag ions and stabilizing the produced Ag nanoparticles.

The main chemical compositions of the chestnut shell extract have been further determined by LC-MS/MS in this study. Thirteen compounds, including catechin, procyanidin, cannabinol, palmitic acid, maslinic acid, oleanolic acid, asiatic acid, L-arginine, scopolamine, sucrose, gentisic acid and 4,5-dicaffeoylquinic acid, have been identified. The crude plant extracts are rich in secondary metabolites such as phenolic acid, polyphenols, alkaloids, tannin, saponin and flavonoids. These bioactive compounds can reduce the Ag^+^ to Ag^0^ in the redox reaction to form the Ag nanoparticles as revealed by recent studies [39,40,41,42]. They further serve as the capping agent on the surface of synthesized nanoparticles to stabilize the nanoparticles. In this study, the polyphenolic compounds, such as catechin, procyanidin and cannabinol, and organic acids such as palmitic acid, maslinic acid, oleanolic acid, asiatic acid, can reduce Ag^+^ to Ag^0^ in the redox reaction to form Ag nanoparticles

In addition, it is interesting that the size of the functionalized chestnut-shell-extract Ag nanoparticles in this study are monodisperse. It has been revealed that the nanomaterials synthesized from a plant extract are usually multi-disperse in most cases because of the complicated components and reactions [43,44]. One possible reason for the formation of monodisperse nanoparticles in the present study is that some components of the chestnut shell extract have a high affinity with Ag ions, resulting in the homogeneous nucleation and growth of Ag nanoparticles. The main components of the chestnut shell extract such as catechin and organic acid may dominate the formation of crystal nuclei and their growth since they have high affinity for Ag^+^. For example, catechin can be used to synthesize uniform Ag nanoparticles [45]. The detailed investigation of the formation mechanism of Ag nanoparticles and their further applications is underway.

In addition to their use in the synthesis of Ag nanoparticles, plant extracts were demonstrated to biosynthesize nano-scale oxides with antibacterial response against *E. coli* and *S. aurus* [46,47,48]. Furthermore, plant extracts can be utilized for the green synthesis of nano-sized oxides with superior physical and electrochemical properties [49,50,51,52]. Based on our results with Ag nanoparticles with the outstanding quasi-monodispersity, it is expected that chestnut shell extract can be further explored to biosynthesize nano-scaled oxides (such as CuO, ZnO, Fe_2_O_3_ and NiO) with a superior sized quasi-monodispersity.

## 4. Conclusions

Chestnut shell extract has been utilized to synthesize monodisperse crystalline Ag nanoparticles. The obtained functionalized chestnut-shell-extract Ag nanoparticles have been characterized by UV-Vis, FTIR, TEM, XRD and XPS. These measurement results indicate that the obtained nanomaterial is spherical Ag nanoparticles capped by chestnut shell extract, with an average size of 8.82 nm and a face-centered cubic crystal structure. The antibacterial activity test revealed that the Ag nanoparticles modified by the chestnut shell extract had an obvious inhibitory effect on *Escherichia coli*, *Staphylococcus aureus* and *Candida albicans*. Further applications of these novel Ag nanoparticles are underway. In addition, the method developed in this study has facile and eco-friendly characteristics and the potential to be used for the synthesis of some other metal nanoparticles.

## Figures and Tables

**Figure 1 antibiotics-12-00201-f001:**
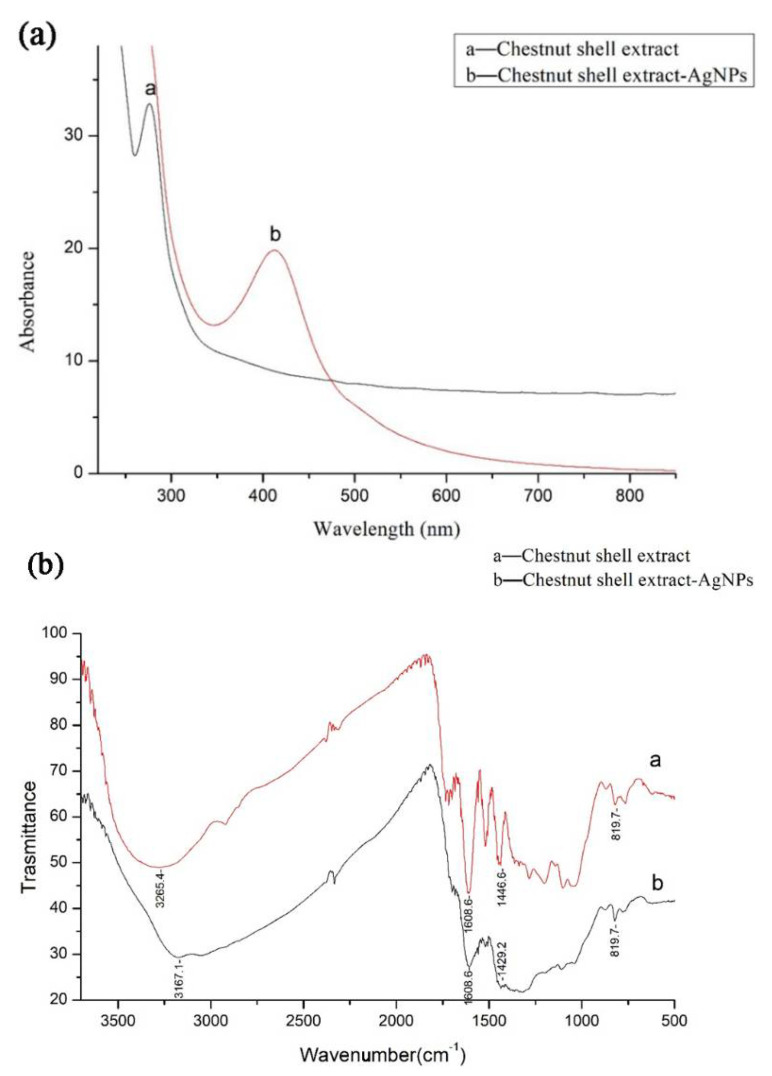
(**a**) The typical UV absorption spectrum of the functionalized chestnut-shell-extract AgNPs (chestnut shell extract solution was also measured for comparison) (**b**) FTIR spectra of the functionalized chestnut-shell-extract AgNPs (chestnut shell extract was also measured for comparison).

**Figure 2 antibiotics-12-00201-f002:**
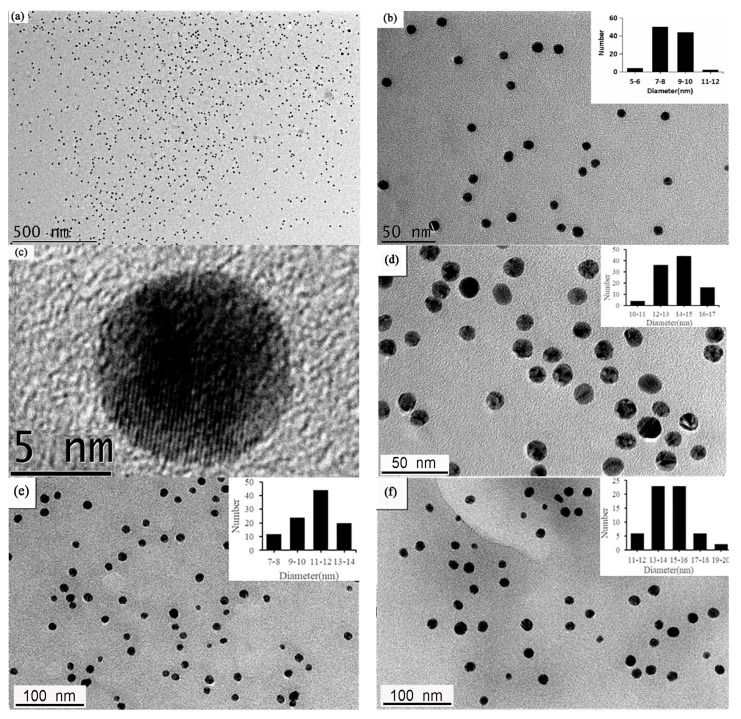
(**a**) Low-magnification TEM image of the functionalized chestnut-shell-extract AgNPs. (**b**) An enlarged TEM image of the Ag nanoparticles. (**c**) High-resolution TEM image of one Ag nanoparticle. (**d**) The representative TEM image of the Ag nanoparticles produced by seed-growth method. (**e**) The TEM image of the synthesized AgNPs at pH 7.0. (**f**) The TEM image of the synthesized AgNPs using 0.8 mL 5% AgNO_3_ solution.

**Figure 3 antibiotics-12-00201-f003:**
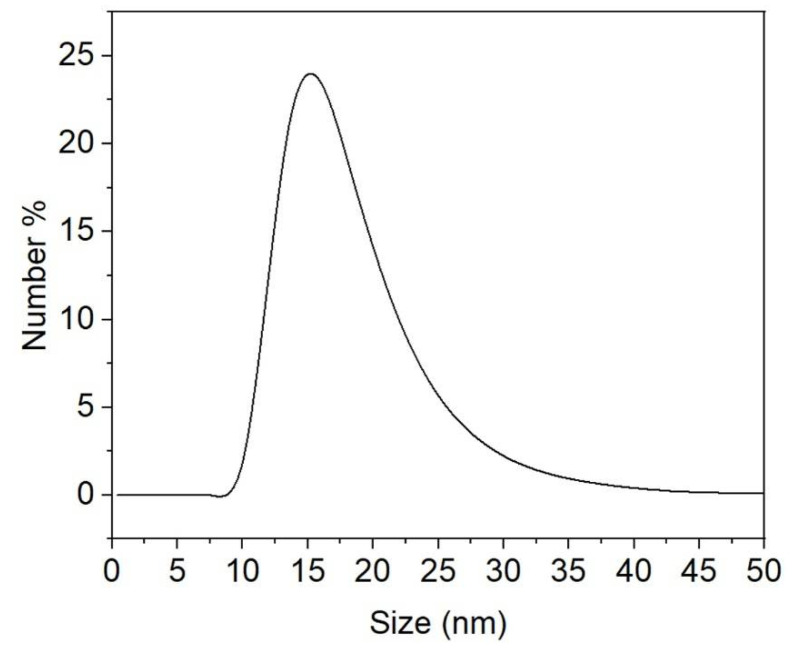
The size distribution of the functionalized chestnut-shell-extract AgNPs measured by DLS.

**Figure 4 antibiotics-12-00201-f004:**
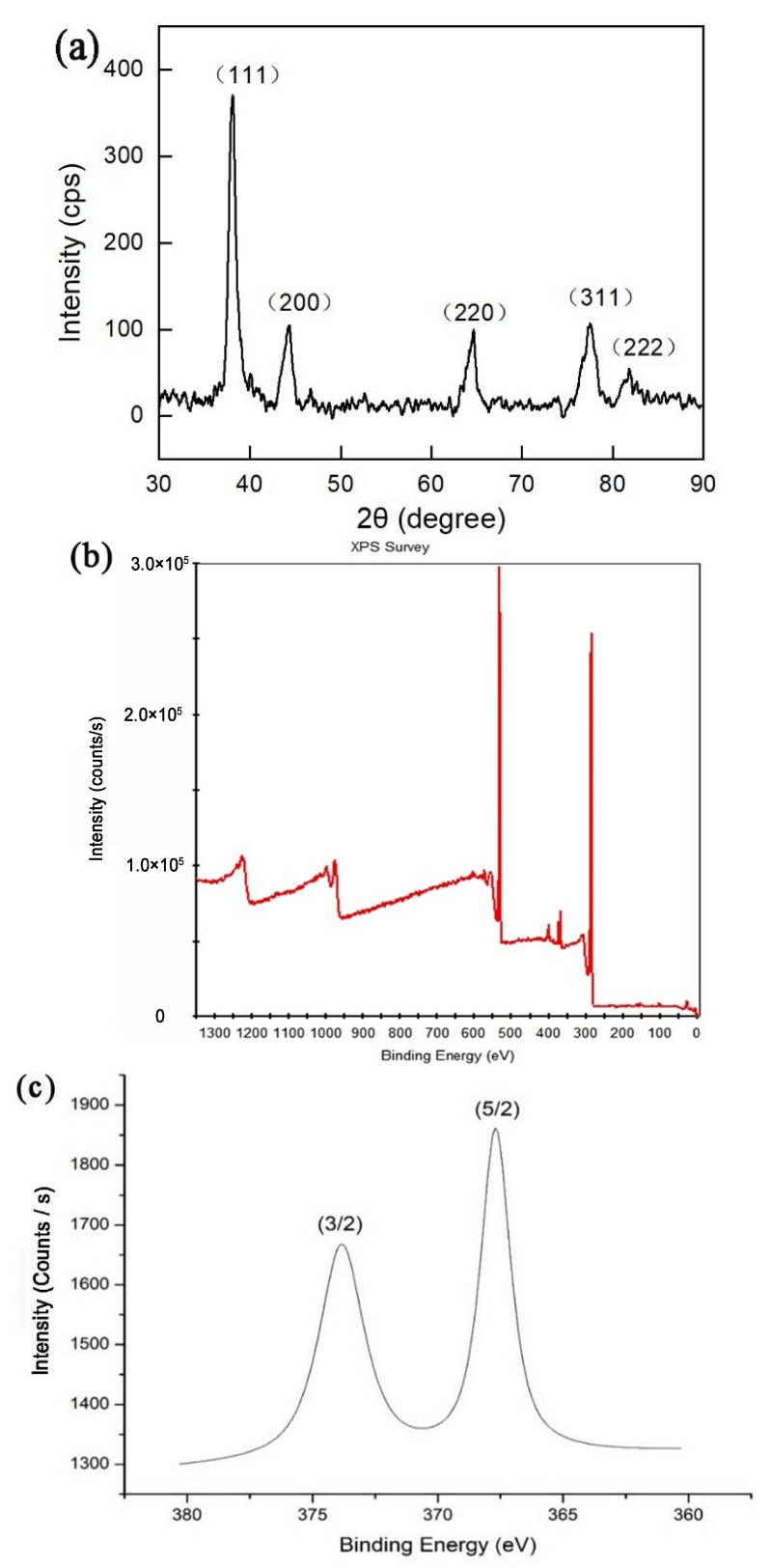
(**a**) XRD spectrum of the functionalized chestnut-shell-extract Ag nanoparticles. (**b**) XPS pattern of the functionalized chestnut-shell-extract AgNPs. (**c**) XPS pattern of Ag element in the functionalized chestnut-shell-extract AgNPs.

**Figure 5 antibiotics-12-00201-f005:**
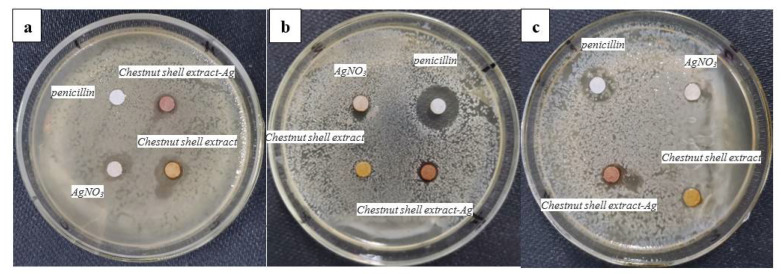
The photograph of the zone of inhibition of the functionalized chestnut-shell-extract Ag nanoparticles, the chestnut shell extract, AgNO_3_ solution and penicillin against (**a**) *Escherichia coli* (**b**) *Candida albicans* (**c**) *Staphylococcus aureus*.

**Figure 6 antibiotics-12-00201-f006:**
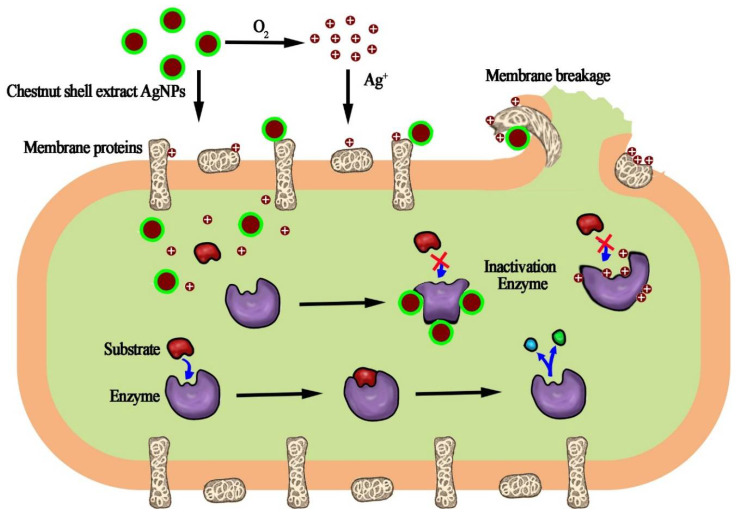
Schematic illustration of the possible antimicrobial mechanism of the functionalized chestnut-shell-extract AgNPs.

**Table 1 antibiotics-12-00201-t001:** The MIC and MBC of functionalized chestnut-shell-extract AgNPs and control samples.

μg/mL	*E. coli*		*S. aureus*		*C. albicans*	
	MIC	MBC	MIC	MBC	MIC	MBC
Chestnut-shell-extract AgNPs	0.13	0.25	0.25	0.50	0.13	0.25
AgNPs from NaBH_4_ reduction	21.25	42.50	21.25	42.50	21.25	42.50
Penicillin	1.60	3.30	6.00	12.00	6.00	12.00

## Data Availability

Data are available from the corresponding author upon reasonable request.

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
