# Peer review of "Green Synthesis and Antimicrobial Study on Functionalized Chestnut-Shell-Extract Ag Nanoparticles"

_antibiotics, 2023, doi:10.3390/antibiotics12020201_

Round 1
Reviewer 1 Report
The present work is on the green synthesis and antimicrobial investigation on chestnut shell extract-mediated Ag nanoparticles. The reported findings are attractive, but similar work is already published (Shao, Kai, and Sun, Jixiang and Sun, Congcong and Wang, Xitong and Lin, Yamei and Zhi, Hongxin and Fu, Yujie and liu, zhiguo, Green Synthesis of the Monodisperse Crystalline Ag Nanoparticles Using Chestnut Shell Extract. Available at SSRN: https://ssrn.com/abstract=4087341 or http://dx.doi.org/10.2139/ssrn.4087341.)
I, therefore, feel that this manuscript could not be considered for the said journal.
Author Response
Response: The article mentioned by reviewer is the preprint of the present work. We really did not know that preprint version until check the preprint link given by the reviewer. We further checked the preprint information and concluded that this preprint is derived from the submission to an Elsevier journal before to antibiotics. We may wrong check the box about “I want to share my research early and openly as a preprint” Nevertheless, we decide to remove this preprint from SSRN. After communication with the SSRN, the preprint has been removed from SSRN by our request. Thanks very much for reviewer’s help.
Reviewer 2 Report
The contribution is original & sound indeed. Yet a variety of studies on the biosynthesis of nano-scaled Ag but not using chestnut natural extract. In addition, the originality of such a study that it reports quasi mono-disperse Ag NPs. The reported results are sound , comprehensive & complementary in support of the various discussions within the manuscript and the drawn conclusions. Hence it is recommended for publication once the following Precisions (P), Corrections (C) & Recommendations (R) are addressed each & all:
PCR-1:
It would have been sound to highlight the major phyto-compounds within the chestnut extract as they are likely to be at the origin of the Ag salt reduction.
PCR-2:
While the various chracterization studies confirm the bio-synthesis of nanoscaled Ag , the mechanism of their formation has been completely ommitted (the authors vaguely tried to discuss it in section3.3. but it is not enough . The authors should/ are required to highlight the likely potential mechanism of Ag salt reduction by the chestnut phytocompounds. In this regard, the authors are encouraged to refer to the following publications that they should include in their discussion and add them in their reference section:
(i)Biogenic synthesis and antibacterial activity of controlled silver nanoparticles using an extract of Gongronema Latifolium, SO Aisida, K Ugwu, PA Akpa, AC Nwanya, PM Ejikeme, S Botha, I Ahmad, ... Materials Chemistry and Physics 237, 121859(2019)
(ii)Bioreduction potentials of dried root of Zingiber officinale for a simple green synthesis of silver nanoparticles: antibacterial studies, JJ Vijaya, N Jayaprakash, K Kombaiah, K Kaviyarasu, LJ Kennedy, Journal of Photochemistry and Photobiology B: Biology 177, 62-68(2017)
(iii) Green synthesis of nickel oxide, palladium and palladium oxide synthesized via Aspalathus linearis natural extracts: physical properties & mechanism of formation, N Mayedwa, N Mongwaketsi, S Khamlich, K Kaviyarasu, N Matinise, ... Applied Surface Science 446, 266-272(2018)
(iv)Biosynthesis of silver nanoparticles using bitter leave (Veronica amygdalina) for antibacterial activities, SO Aisida, K Ugwu, PA Akpa, AC Nwanya, U Nwankwo, SS Botha, ... Surfaces and Interfaces 17, 100359(2019)
PCR-3:
Yet, the antibacterial response seems effective indeed, it would be wise to compare it to other bio-engineered nanoparticles including that of nano-scale oxides. In this regard, the authors are encouraged to refer to the following publications that they should include in their discussion and add them in their reference section:
(i)Ditlopo, N., Sintwa, N., Khamlich, S. et al. From Khoi-San indigenous knowledge to bioengineered CeO2 nanocrystals to exceptional UV-blocking green nanocosmetics. Sci Rep 12, 3468 (2022). https://doi.org/10.1038/s41598-022-06828-x,
(ii)Havenga, D., Akoba, R., Menzi, L. et al. From Himba indigenous knowledge to engineered Fe2O3 UV-blocking green nanocosmetics. Sci Rep 12, 2259 (2022). https://doi.org/10.1038/s41598-021-04663-0,
(iii)Eco-friendly synthesis, characterization, in vitro and in vivo anti-inflammatory activity of silver nanoparticle-mediated Selaginella myosurus aqueous extract
PBE Kedi, FE Meva, L Kotsedi, EL Nguemfo, CB Zangueu, AA Ntoumba, ... International journal of nanomedicine 13, 8537(2018)
PCR-4:
What is the origin of the quasi-monodispersity of the Ag Nanoparticles obtained via the chestnut natural extract?
PCR-5:
In view of the outstanding quasi-monodispersity of the obtained Ag nanoparticles, is it possible to biosynthesize nanoscaled oxides (such as CuO, ZnO, Fe2O3, NiO,.....)with such a superior size quasi-monodispersity?. The authors are encouraged to include such a discussion at the end of the manuscript just before the conclusion section. This foresight discussion would shedlight & encourage the nano-community to pursue such a study with Chestnut natural extract. In this regard, the authors are encouraged to refer to the following publications that they should include in their discussion and add them in their reference section:
(i)Stalling behaviour of chloride ions: a non-enzymatic electrochemical detection of α-Endosulfan using CuO interface
SS Rathnakumar, K Noluthando, AJ Kulandaiswamy, JBB Rayappan, ...
Sensors and Actuators B: Chemical 293, 100-106(2019)
(ii)Industrial textile effluent treatment and antibacterial effectiveness of Zea mays L. Dry husk mediated bio-synthesized copper oxide nanoparticles
AC Nwanya, LC Razanamahandry, AKH Bashir, CO Ikpo, SC Nwanya, ... Journal of hazardous materials 375, 281-289(2019) (iii)Optical limiting in pulsed laser deposited VO2 nanostructures M Maaza, D Hamidi, A Simo, T Kerdja, AK Chaudhary, JBK Kana Optics Communications 285 (6), 1190-1193(2012) (iv)Physical & electrochemical properties of green synthesized bunsenite NiO nanoparticles via Callistemon viminalis’ extracts BT Sone, XG Fuku, M Maaza Int. J. Electrochem. Sci 11, 8204-8220(2016)
Author Response
PCR-1:
It would have been sound to highlight the major phyto-compounds within the chestnut extract as they are likely to be at the origin of the Ag salt reduction.
Response: The context of the major phyto-compounds within the chestnut extract has been added in the revised manuscript according to the reviewer’s valuable comments.
While the various chracterization studies confirm the bio-synthesis of nanoscaled Ag , the mechanism of their formation has been completely ommitted (the authors vaguely tried to discuss it in section3.3. but it is not enough . The authors should/ are required to highlight the likely potential mechanism of Ag salt reduction by the chestnut phytocompounds. In this regard, the authors are encouraged to refer to the following publications that they should include in their discussion and add them in their reference section:
(i)Biogenic synthesis and antibacterial activity of controlled silver nanoparticles using an extract of Gongronema Latifolium, SO Aisida, K Ugwu, PA Akpa, AC Nwanya, PM Ejikeme, S Botha, I Ahmad, ... Materials Chemistry and Physics 237, 121859(2019)
(ii)Bioreduction potentials of dried root of Zingiber officinale for a simple green synthesis of silver nanoparticles: antibacterial studies, JJ Vijaya, N Jayaprakash, K Kombaiah, K Kaviyarasu, LJ Kennedy, Journal of Photochemistry and Photobiology B: Biology 177, 62-68(2017)
(iii) Green synthesis of nickel oxide, palladium and palladium oxide synthesized via Aspalathus linearis natural extracts: physical properties & mechanism of formation, N Mayedwa, N Mongwaketsi, S Khamlich, K Kaviyarasu, N Matinise, ... Applied Surface Science 446, 266-272(2018)
(iv)Biosynthesis of silver nanoparticles using bitter leave (Veronica amygdalina) for antibacterial activities, SO Aisida, K Ugwu, PA Akpa, AC Nwanya, U Nwankwo, SS Botha, ... Surfaces and Interfaces 17, 100359(2019)
Response: Thanks very much for reviewer’s valuable comments. We have added the discussion about the potential mechanism of Ag salt reduction by the chestnut phytocompounds in the revised manuscript according to the reviewer’s comments. These excellent literatures mentioned by reviewer have been cited in the revised manuscript.
PCR-3:
Yet, the antibacterial response seems effective indeed, it would be wise to compare it to other bio-engineered nanoparticles including that of nano-scale oxides. In this regard, the authors are encouraged to refer to the following publications that they should include in their discussion and add them in their reference section:
(i)Ditlopo, N., Sintwa, N., Khamlich, S. et al. From Khoi-San indigenous knowledge to bioengineered CeO2 nanocrystals to exceptional UV-blocking green nanocosmetics. Sci Rep 12, 3468 (2022). https://doi.org/10.1038/s41598-022-06828-x,
(ii)Havenga, D., Akoba, R., Menzi, L. et al. From Himba indigenous knowledge to engineered Fe2O3 UV-blocking green nanocosmetics. Sci Rep 12, 2259 (2022). https://doi.org/10.1038/s41598-021-04663-0,
(iii)Eco-friendly synthesis, characterization, in vitro and in vivo anti-inflammatory activity of silver nanoparticle-mediated Selaginella myosurus aqueous extract
PBE Kedi, FE Meva, L Kotsedi, EL Nguemfo, CB Zangueu, AA Ntoumba, ... International journal of nanomedicine 13, 8537(2018)
Response: We have added a discussion according to reviewer’s suggestion. These excellent literatures have been cited in the revised manuscript.
PCR-4:
What is the origin of the quasi-monodispersity of the Ag Nanoparticles obtained via the chestnut natural extract?
Response: We have added a discussion according to reviewer’s comments.
PCR-5:
In view of the outstanding quasi-monodispersity of the obtained Ag nanoparticles, is it possible to biosynthesize nanoscaled oxides (such as CuO, ZnO, Fe2O3, NiO,.....)with such a superior size quasi-monodispersity?. The authors are encouraged to include such a discussion at the end of the manuscript just before the conclusion section. This foresight discussion would shedlight & encourage the nano-community to pursue such a study with Chestnut natural extract. In this regard, the authors are encouraged to refer to the following publications that they should include in their discussion and add them in their reference section:
(i)Stalling behaviour of chloride ions: a non-enzymatic electrochemical detection of α-Endosulfan using CuO interface
SS Rathnakumar, K Noluthando, AJ Kulandaiswamy, JBB Rayappan, ...
Sensors and Actuators B: Chemical 293, 100-106(2019)
(ii)Industrial textile effluent treatment and antibacterial effectiveness of Zea mays L. Dry husk mediated bio-synthesized copper oxide nanoparticles
AC Nwanya, LC Razanamahandry, AKH Bashir, CO Ikpo, SC Nwanya, ... Journal of hazardous materials 375, 281-289(2019) (iii)Optical limiting in pulsed laser deposited VO2 nanostructures M Maaza, D Hamidi, A Simo, T Kerdja, AK Chaudhary, JBK Kana Optics Communications 285 (6), 1190-1193(2012) (iv)Physical & electrochemical properties of green synthesized bunsenite NiO nanoparticles via Callistemon viminalis’ extracts BT Sone, XG Fuku, M Maaza Int. J. Electrochem. Sci 11, 8204-8220(2016)
Response: Thanks very much for reviewer’s help. We have added a discussion at the end of the manuscript just before the conclusion section according to reviewer’s valuable suggestions. These excellent literatures have been cited in the revised manuscript.
Reviewer 3 Report
The manuscript contains valuable information. However, it needs some modifications as follows:
1. The DLS analysis is suggested to be included to determine the hydrodynamic size of synthesized silver nanoparticles.
2. Zeta potential analysis is suggested to be included to show the electrostatic stability of colloidal silver nanoparticles.
3. The authors are suggested to include a schematic figure representing the proposed antibacterial activity of silver nanoparticles.
4. The section of Discussion needs modification. The authors are suggested to compare their findings of the antimicrobial properties of silver nanoparticles with similar studies. The authors should also mention the factors that may have an influence on the biological activity of inorganic nanoparticles. These factors include size distribution, morphology, surface charge, surface chemistry, capping agents, etc. Follow and cite the article below to support the above explanations. https://doi.org/10.1016/j.inoche.2021.108647
5. Follow and cite the following articles in appropriate places:
Penicillium chrysogenum-Derived Silver Nanoparticles: Exploration of Their Antibacterial and Biofilm Inhibitory Activity Against the Standard and Pathogenic Acinetobacter baumannii Compared to Tetracycline
Bioengineering of green-synthesized silver nanoparticles: In vitro physicochemical, antibacterial, biofilm inhibitory, anticoagulant, and antioxidant performance
Biogenic metal nanomaterials to combat antimicrobial resistance
Author Response
The manuscript contains valuable information. However, it needs some modifications as follows:
- The DLS analysis is suggested to be included to determine the hydrodynamic size of synthesized silver nanoparticles.
Response: The DLS analysis has been done and added in the revised manuscript according to the reviewer’s valuable comments.
- Zeta potential analysis is suggested to be included to show the electrostatic stability of colloidal silver nanoparticles.
Response: Zeta potential analysis has been added in the revised manuscript according to the reviewer’s valuable comments.
- The authors are suggested to include a schematic figure representing the proposed antibacterial activity of silver nanoparticles.
Response: A schematic Figure representing the proposed antibacterial activity of silver nanoparticles has been added in the revise manuscript according to the reviewer’s valuable comments.
- The section of Discussion needs modification. The authors are suggested to compare their findings of the antimicrobial properties of silver nanoparticles with similar studies. The authors should also mention the factors that may have an influence on the biological activity of inorganic nanoparticles. These factors include size distribution, morphology, surface charge, surface chemistry, capping agents, etc. Follow and cite the article below to support the above explanations. https://doi.org/10.1016/j.inoche.2021.108647
Response: The comparison of the antimicrobial properties of silver nanoparticles with similar studies and a discussion about the factors that may have an influence on the biological activity has been added in the revised manuscript. The excellent article mentioned by reviewer has been cited in the revised manuscript.
- Follow and cite the following articles in appropriate places:
Penicillium chrysogenum-Derived Silver Nanoparticles: Exploration of Their Antibacterial and Biofilm Inhibitory Activity Against the Standard and Pathogenic Acinetobacter baumannii Compared to Tetracycline
Bioengineering of green-synthesized silver nanoparticles: In vitro physicochemical, antibacterial, biofilm inhibitory, anticoagulant, and antioxidant performance
Biogenic metal nanomaterials to combat antimicrobial resistance
Response: These excellent articles mentioned by reviewer has been cited in the revised manuscript. Thanks very much for your help!
Round 2
Reviewer 1 Report
The present work is on green synthesis and antimicrobial investigation on chestnut shell extract-mediated Ag nanoparticles. The reported findings are attractive, but similar work is already published and indexed in google scholar searching (Shao, Kai and Sun, Jixiang and Sun, Congcong and Wang, Xitong and Lin, Yamei and Zhi, Hongxin and Fu, Yujie and liu, zhiguo, Green Synthesis of the Monodisperse Crystalline Ag Nanoparticles Using Chestnut Shell Extract. Available at SSRN: https://ssrn.com/abstract=4087341 or http://dx.doi.org/10.2139/ssrn.4087341.)
I, therefore, feel that this manuscript could not be considered for the said journal.
